

# Evaluation oxygen deficiency in the Chesapeake Bay

Wencheng L. Slater[1], James. J. Pierson[1], Michael R. Roman[1]

[1]Horn Point Laboratory, University of Maryland Canter of Environmental Science

*Correspondence to*: Wencheng L. Slater (kliu@umces.edu)

5 **Abstract.** The objectives of the Dead Zone Zooplankton research project (NSF OCE-0961942) were to study the effects of the onset, development, and dissipation of hypoxia (DO $< 2$ mg L$^{-1}$) on the plankton food web in the Chesapeake Bay. Here, we present the hydrologic and meteorology data from CTD, Scanfish, and the Safety Measurement System (SMS) of the research vessel to describe the environmental conditions of the bay during the project. We collected data from the mesohaline portion of Chesapeake Bay from 37.5–38.5 °N and from 76–76.5 °W during six research cruises in 2010 and 10 2011. We analyzed the temperature, salinity, and dissolved oxygen from hourly CTD casts using principal component analysis (PCA) to understand variations in hydrography among depths, stations, seasons, and years. In addition to using the commonly accepted standard of hypoxia (DO $< 2$mg L$^{-1}$), we also estimated the oxygen supply and demand of the copepod *Acartia tonsa* according to the surrounding temperature and salinity. The hypoxia in the bay began in the late spring, developed from the bottom layer upstream, progressed toward the sea, became fully established in summer, and gradually 15 dissipated in autumn beginning in the downstream regions. However, we observed that extreme weather could interrupt this succession and reignite hypoxia events after summer. Our PCA results indicated that temperature was the major driver of environmental conditions, and dissolved oxygen in the bottom layer was the second most important driver. Within each temperature group, we found that samples from 2011 and the north station were less oxygenated than samples from 2010 and the south station. Comparing the two metrics of oxygen deficiency, we found that the duration, distribution, and severity of 20 environmental oxygen deficiency could be underestimated using the traditional metric, especially under warm and salty conditions. We recommend that temperature and species-specific metrics be considered along with dissolved oxygen concentration when setting water quality goals for management. We uploaded the CTD data to the Biological and Chemical Oceanography Data Management Office (DOI:10.1575/1912/bco-dmo.687991; hyphen is part of the DOI), and we stored the Scanfish and SMS system data in the Rolling Deck to Repository (DOI: 10.7284/901570; 10.7284/901574; 10.7284/907638; 25 10.7284/901618; 10.7284/902443; 10.7284/902721 for the six cruises, respectively).

## 1 Introduction

### 1.1 Hypoxia in the Bay

The amount of oxygen-deficient water (dissolved oxygen $< 2$ mg L$^{-1}$) in coastal areas has been increasing worldwide in the recent decades largely due to eutrophication and warming (Rabalais et al., 2010; Rhein et al., 2013). Reported from more 30 than 400 systems and 245,000 km$^2$ area, hypoxia has evolved into a key stressor in many aquatic ecosystems, especially in those populated and developed regions (Diaz and Rosenberg, 2008). As the largest estuary in the United States, the Chesapeake Bay is prone to hypoxia due to its seasonal flushing which strengthens stratification and its linear shape that impedes full circulation (Libes, 2011). Besides natural causes, anthropogenic causes like eutrophication and warming have contributed to hypoxia in the Bay as well (Cowan and Boynton, 1996). Chesapeake Bay has an area of 3500 km$^2$ and 35 annually a three-month duration of deoxygenated bottom water. The volume of hypoxic water has been increasing, and the onset timing has been shifting earlier since the 1950s (Boesch et al., 2001). With both temperature and human population (the major source of eutrophication) projected to increase, hypoxic volume in the Bay will likely increase (Deutsch et al., 2011; Najjar et al., 2010; Rabalais et al., 2009).



The consequences of hypoxia are both economic and environmental. Hypoxia can have large impacts on fisheries by decreasing landings, and losses due to hypoxia were reported in many ecosystems, including Mobile Bay's oysters fisheries (*Crassostrea virginica*), North Carolina's brown shrimp (*Farfantepenaeus aztecus*) fisheries, and the Black Sea's Norway

Lobster (*Nephrops norvegicus*) fisheries (summarized in Noone, Diaz, & Sumaila, 2013). A 10-year study of Chesapeake Bay also indicated that chronic hypoxia in the Chesapeake Bay reduced the number and catch rates of demersal fish species on a large scale (Buchheister et al., 2013). Lipton & Hicks (2003) estimated US$200,000 net losses of value to the recreational striped bass (*Morone saxatilis*) fishery in the Patuxtant River due to hypoxia, and the losses would be > US$145 million if projected to the whole bay. Although it is difficult to quantify exact economical loss due to hypoxia, most studies

agreed, at least, fishery landing decreased with oxygen deficiency. The environmental changes under hypoxic conditions may be systematic. For example, hypoxia could alter foodweb dynamics of an ecosystem by removing sensitive species such as fast-swimming fish and advantaging tolerate species like slow-drifting jellyfish (Breitburg et al. 1994). Bottom hypoxia also causes habitat degradation and leads to mass mortality of benthos and fish (reviewed in Diaz & Rosenberg, 1995). It was proposed that K-selected species would be replaced by r-selected species and a complex foodweb by a simpler foodweb

(reviewed in Wu, 2002).

### 1.2 Not all hypoxia is the same

While most species in the Bay are not immune from the effects of hypoxia, not every specie experiences hypoxia in the same way. Fast swimming fish demand more oxygen than slow drifting zooplankton, and therefore a striped bass requires more dissolved oxygen than a jellyfish. Even for the same species, oxygen demand is higher under warmer conditions because

metabolism increases with temperature. Therefore, 2 mg $L^{-1}$ might not be a universal standard for all species under all conditions. Elliott et al. (2013a, 2013b) analyzed the Bay hypoxia for the dominant copepod *Acartia tonsa* with its lowest sustainable (basal) respiration rate, the oxygen-independent (target) respiration rate, and the surrounding partial pressure of dissolved oxygen ($pO_2$). When the surrounding $pO_2$ is enough to support the target respiration rate ($pO_2 \geq P_{crit}$), the copepod respiration rate is stable and independent from the surround $pO_2$. However when $pO_2 < P_{crit}$, the copepod respiration rate

linearly declines with $pO_2$ and copepods may suffer sublethal effects. If $pO_2$ drops to the point that it no longer meets copepod's basal respiration needs ($P_{leth}$), copepod mortality is expected. In this study we also compare the oxygen supplies ($pO_2$) with *A. tonsa*'s oxygen needs ($P_{crit}$, $P_{leth}$) to re-categorize the "dead zone" with biological standards.

Since the effects of hypoxia are systematic in the Bay, dissolved oxygen has been adopted as a key water quality indicator in

many environment management and conservation plans. To provide information for water quality management and better prepare for a changing environment due to climate change and anthropogenic stressors, we researched hypoxia and its interactions with different environmental conditions. The overarching goal of this paper is to understand conditions of oxygen deficiency in the Bay during our fieldwork in 2010 and 2011. The objectives of this paper are to 1) describe the development of seasonal hypoxia and hurricane-influenced hypoxia in the Chesapeake Bay mainstream and 2) to compare

areas and durations of hypoxia events with different metrics (dissolved oxygen concentration vs. oxygen supply and demand).

## 2 Methods and Materials

### 2.1 Data collecting and analysis

We conducted six, week-long cruises with the R/V Hugh R. Sharp in the main channel of the Chesapeake Bay from late

spring to autumn (May, July/August, and September) in 2010 and 2011. We began each cruise with a Scanfish survey (~79 nautical miles) along the main channel of the Bay from approximately the south of Bay Bridge (39.00N, 76.37 W) to the



Rappahannock Shoals (37.70 N, 76.19 W) (Fig 1). Following the survey, we anchored at two stations, termed North (38° 31.32' N, 076° 24.48' W) and South (37° 43.68' N, 076° 12.0' W) (Fig 1), to collect abiotic and biotic data. We spent approximately 2.5 days at each station, with ~27 hours at anchor and ~33 hours trawling nearby the station with zooplankton and fishnets.

Scanfish surveys were also conducted while transitioning from the South station toward north, to a point approximately 15km northern of the North and then returned to the North, except in May 2011 we conducted CTD survey along the Bay instead of Scanfish due to mechanical problems (Fig 1d). In addition to the regular survey path, we extended the Scanfish survey to the upper bay during the 2011 Autumn cruise to observe the post-hurricane conditions (Fig 1f). More cruise and

sampling details were described at Pierson et al. (2017). During each cruise, meteorology and sea surface data were collected every 10 minutes with the Surface Mapping System (SMS) on the R/V Sharp during the cruises. The CTD datasets were submitted to the Biological and Chemical Oceanography Data Management Office (BCO-DMO) at https://hdl.handle.net/1912/8933 (DOI: 10.1575/1912/bco-dmo.687991; hyphen is part of the DOI) and the Scanfish and meteorology data from the SMS system were uploaded to RVdata.us (DOI: 10.7284/901570; 10.7284/901574;

10.7284/907638; 10.7284/901618; 10.7284/902443; 10.7284/902721). While at anchor, we conducted hourly CTD casts to obtain temperature, salinity, dissolved oxygen, and fluorescence data at 0.5 m depth intervals.

Both the Scanfish and SMS data were processed in Matlab, and CTD data were first processed in Sea-Bird data processing software and then in Matlab (details described in Pierson et al. 2017). For each CTD cast, pycnocline depths were chosen as

the depth of maximum density gradient. Accordingly, we defined three water layers: above, at, and below the pycnocline, and then we averaged temperature, salinity, and dissolved oxygen within each layer for each cast. These nine variables from each CTD cast were further sorted with Principle Component Analysis (PCA) in R for grouping similar cruises and stations. Based on the PCA results, we grouped similar cruises and stations according to their distributions on the scatter plot comparing the first principle component (PC1) to the second principle component (PC2). Data points were first grouped by

their PC1 scores, and then divided into subgroups by their PC2 scores.

### 2.2 Oxygen supplies and needs

We estimated whether the temperature-specific oxygen demands were met for the copepod *A. tonsa* at each data point. First, we calculated $Q_{10}$ of *A. tonsa* with salinity (Eq. 1, explained in Elliott, Pierson, & Roman, 2013b) and oxygen solubility (O2Sat, Weiss, 1970) with the "sw_satO2.m" in the SeaWater MATLAB toolbox. From oxygen solubility, we calculated the

percentage oxygen saturation ($O_2P_{ct}$, Eq. 2) and then the saturation partial pressure of environmental oxygen ($p_{O2}$, Eq. 3). From $Q_{10}$ and temperature, we estimated the temperature-specific critical oxygen partial pressure ($P_{crit}$, Eq. 4) and the lethal oxygen partial pressure ($P_{leth}$, Eq. 5) (Elliott et al., 2013b; Pierson et al., 2017). By comparing $pO_2$ with $p_{crit}$ and $P_{leth}$, we estimated the oxygen supply and the oxygen demands of copepod *A. tonsa* under different temperature and salinity conditions. If $pO_2 > P_{crit}$, the metabolism of copepod *A. tonsa* is independent of the surrounding $pO_2$. If $pO_2 < P_{crit}$

(biological hypoxia), the metabolism of copepod *A. tonsa* decreased with $pO_2$, and the copepod was likely suffered from sublethal effects. If $pO_2 < P_{leth}$, the surrounding dissolved oxygen was not enough to support the minimum survivable respiration and hypoxia-induced mortality would increase.

$$Q_{10} = 0.053 \times Salinity + 0.705 \qquad (1)$$

$$O_2Pct\ (\%) = \frac{DO}{O_2Sat} \qquad (2)$$

$$pO_2 = (159.27 \times O_2Pct - 0.0141) \times 133.322368 \qquad / \qquad (3)$$



*1000*

$$P_{crit} = 7.49Q_{10}{}^{0.1(T-18)} + 0.59 \qquad (4)$$

$$P_{leth} = 2.61Q_{10}{}^{0.1(T-18)} + 0.59 \qquad (5)$$

## 3 Results

### 3.1 Environmental conditions and hypoxia in the Bay

We conducted in total 229 and 223 hourly CTD casts during 2010 and 2011 cruises (Table 1), respectively. We observed that temperature, salinity, dissolved oxygen, and chlorophyll varied largely with water layers, stations, seasons, and years. The
CTD (Fig 2, 3), Scanfish (Fig 4 - 7), and SMS records (Fig 8 - 10) all indicated the North station had lower salinity on the surface and less dissolved oxygen in the bottom than the South station, and 2011 had lower salinity and less dissolved oxygen than 2010. The results from the CTD casts, Scanfish surveys, and the SMS records showed similar hydrographic trends: The highest water surface temperature (~ 31.6°C) was recorded during the 2011-Summer cruise and lowest (~ 11.2°C ) during the 2011-Spring cruise (Fig 2, 4, 8); the highest salinity (~ 29.7) was recorded during the 2010-Autumn cruise and
lowest ( ~ 0.2) during the 2011-Spring cruise (Fig 2, 5, 8); the highest dissolved oxygen (15.3 mg L$^{-1}$) was recorded during the 2011-Spring cruise and lowest (below detectable limits) during the 2010-Summer and 2011-Spring and Summer cruises (Fig 2, 6); the highest chlorophyll-a (24.5 μg L$^{-1}$) was recorded during the 2011-Spring cruise and lowest (1.02 μg L$^{-1}$) during the 2011-Summer cruise (Fig 3, 7, 8). The SMS also observed both the highest (1024.00 bar) and lowest (1007.11 bar) air pressure from the 2010-Spring cruise (Fig 9), and the highest wind speed from the 2011-Spring cruise (29.03 knots)
and lowest the 2010-Autumn cruise (0.80 knots) (Fig 10).

With the CTD and Scanfish results, we found the strength and the distribution of hypoxia in the Chesapeake Bay varied with depths, locations, seasons, years, and significant weather events (Fig 2, 6). According to the commonly adopted standard of hypoxia in coastal studies (DO < 2 mg L$^{-1}$, Nixon, 1995), the hypoxic area of mid-Chesapeake Bay was confined below
pycnoclines during summer. We observed as Kemp et al. (2005) described that the bottom hypoxia developed first in the northern Bay during spring, became established in summer, and dissipated in autumn beginning from the southern Bay (Fig 6). Besides spatial and seasonal variations, we also observed interannual variations. In 2011 stream flow was higher, and the surface salinity was lower than 2010 (Fig 5), which led to stronger stratification and more intense bottom hypoxia in 2011 than in 2010 (Fig 6). During the 2011-Summer cruise, the dissolved oxygen was below the detectable limit at the bottom of
the North station, and the majority part of the water below pycnoclines was either severely hypoxic (DO < 1 mg L$^{-1}$) or anoxic (DO = 0 mg L$^{-1}$). Also, Hurricane Irene passed the Chesapeake Bay two weeks before the 2011 Autumn cruise, and we observed that hypoxia dissipated slower in 2011 than in 2010 (Fig 5c, 5f). Hypoxia almost disappeared during the 2010-Autumn cruise except for the very bottom layer at the North station, while in the 2011-Autumn cruise more than 50% of the vertical water column was still hypoxic at the North station. In general water from the bottom layer, the North station, during
summer, and throughout 2011 were less oxygenated than samples from the surface layer, the South station, non-summer seasons, and throughout 2010, except for after the hurricane.

### 3.2 Principal Component Analysis and Grouping

Only PC1 and PC2 had eigenvalues larger than 1, and together the two principal components explained 82% of the
variability in environmental conditions (Table 2a). PC1 explained 56% of the variance (Table 2a), and the top three loadings were water temperatures of the bottom layer (0.42), pycnocline (0.39), and the surface layer (0.38), indicating water temperature was the major driving factor of this PC (Table 2b). PC2 explained 26% of the variance (Table 2a), and the top



three loadings were dissolved oxygen of bottom layer (0.51) and salinity of the bottom (0.45) and surface layers (0.40) (Table 2b), indicating the bottom dissolved oxygen and salinity were the major drivers of PC2.

A scatter plot of PC1 (x-axis) and PC2 (y-axis) for all hourly CTD casts conducted while at anchor, from both stations and all six cruises, shows that CTD data roughly fell into three groups according to the PC1 scores (Fig 11). Because water temperature and PC1 were positively related (Table 2b), we named the three groups "Cool (C)", "Temperate (T)", and "Warm (W)", from left to right on PC1, corresponding to PC1 scores of −4 - −2, −2 - 0, and 0 - 2, respectively. Within each PC1 group, we further divided the data into two subgroups according to their PC2 scores. Since Bottom dissolved oxygen was the biggest loading on PC2 and they were positively related (Table 2b), we named the PC2 subgroups "Less-
Oxygenated (LO)" and "More-Oxygenated (MO)" from bottom to top on PC2, corresponding roughly to PC2 scores < 0 or > 0, respectively. The cruises and stations and their corresponding groups are listed in Table 3. All spring cruises fell into the C groups, and most of them belong to C-MO subgroup except 2011-Spring, North, where the bottom water column was severely hypoxic (DO close to 0 mg $L^{-1}$ and $pO_2 < p_{leth}$, Fig 11 & Table 3). All summer cruises from both years and the 2010-Autumn cruise fell in the W group. Data from 2010-Autumn (both stations) and 2010-Summer (South station) were grouped
into W-MO, while data from 2011-Summer (both stations) and 2010-Summer (North station) were grouped into W-LO. Only the 2011-Autumn cruises belong to the T group, and the data from the North were in the T-LO group while South in the T-MO group.

    The mean water temperature was 20.01 ± C, 23.02 ± 0.37°C, and 25.39 ± 1.62°C in the C, T, and W group, respectively. The
temperature variations among vertical water layers and between hypoxia subgroups were smaller compared with the variations among groups (Table 4). In general, the W group had higher salinity (14 - 20) than the C (5 - 18) and T (8 - 17) groups (Table 4). This may have been due to strong evaporation under higher temperature in the W group, and both the spring freshet and rainfall during Hurricane Irene in 2011 Autumn that lead to lower salinity in the C and T group. The LO groups displayed lower salinity especially in the surface (5 - 14), since most of the components in this group were North
stations with more freshet input, indicating stronger strobilation. The mean dissolved oxygen was 7.05 ± mg $L^{-1}$, 5.35 ± 2.12 mg $L^{-1}$, and 4.57 ± 2.81° mg $L^{-1}$ in the C, T, and W group, respectively (Table 4). Warmer groups in general had lower averaged DO due to lower oxygen solubility and the stronger stratification in summer which led to the bottom hypoxia. Also, organisms have higher metabolic rates under warmer conditions and therefore consume more dissolved oxygen.

### 3.3 Comparing different metrics for oxygen deficiency

The mean dissolved oxygen was higher in the MO subgroups (6.34 ± 2.90 mg $L^{-1}$) than in LO subgroups (4.60 ± 4.00 mg $L^{-1}$). The standard deviations in the LO groups were larger because the oxygen-deficient water columns were mostly in the bottom of the water column while the surface water was still sufficiently oxygenated. We applied both hydrological (measuring dissolved oxygen concentration) and biological standards (comparing $pO_2$ with *A. tonsa*'s oxygen demands *in situ*) to categorize the "dead zone" in the Chesapeake Bay. For the hydrological standard, we found only the bottom water of
C-LO and W-LO groups met DO < 2 mg $L^{-1}$ (Table 3 & Fig 6), and thus only during those conditions would water be considered hypoxic. However if using the biological standards, we found bottom water of all the LO groups and also the W-MO group met the hypoxic definition for ($pO_2 < p_{crit}$) for *A. tonsa* (Table 4, Fig 2). With the biological hypoxia matrix ($pO_2 < p_{crit}$, Elliott, Pierson, & Roman, 2013b), the comparisons between the hypoxia subgroups in the C and T groups were made between normoxic ($pO_2 > P_{crit}$) and hypoxic conditions ($pO_2 < p_{crit}$), but the comparisons in the W group were made between
moderate ($pO_2 < P_{crit}$) and severe hypoxic ($pO_2 < p_{leth}$) (Table 3). By calculating the gap between oxygen demands (estimated $P_{crit}$ with surrounding temperature and salinity) and supplies (surrounding $pO_2$) in the bottom water column, the observed order of severity of bottom oxygen deficiency was C – LO > W – LO > T – LO > W – MO > C – MO > T – MO (Table 4).





The two hypoxia indicators (DO and $p$O$_2$) mapped different "dead zones" in the Bay. Considering the temperature dependent oxygen demands of copepod *A. tonsa*, the biological hypoxia was actually larger and longer than hydrological hypoxia (DO < 2 mg L$^{-1}$). In both 2010 and 2011, the $p$O$_2$ was less than $p_{crit}$ below the pycnocline at the North station from spring to autumn, and $p$O$_2$ was less than $p_{leth}$ below the pycnocline during summer (Fig 2). In 2011, we observed biological hypoxia from spring to autumn, and the areas of biological hypoxia were bigger than 2010. According to the biological standards, the oxygen-deficient water mass in the Chesapeake Bay was not limited to the bottom water. Instead, hypoxia conditions developed from the bottom water of the northern Bay, spread out to the southern Bay and broke through pycnoclines in summer, and retreated to the northern bay in autumn.

## 4 Discussion

The goal of this project was to understand the succession of hypoxia in the Chesapeake Bay and to compare two different metrics of oxygen deficiency. Our study contained fine resolution hydrographic data including temperature, salinity, dissolved oxygen, fluorescence, and also air temperature, pressure, and wind speeds. Our samples also provided a contrast among depth, stations, seasons, years, and weather conditions. The PCA analysis enabled future studies to compare the effects of temperature and oxygen deficiency with proper grouping. The comparison between two metrics of oxygen deficiency indicated potentially underestimation of the hypoxia severity with the current standard, and we suggest a more holistic consideration of the future water quality management.

### 4.1 Hydrological hypoxia and biological hypoxia

In our study, the low-dissolved oxygen "dead zone" in the mid-Chesapeake Bay was considerably larger and the event duration lasted longer using a biological standard ($p$O$_2$ < $p_{crit}$) than the commonly adopted hydrological standard to define hypoxia (DO < 2mg L$^{-1}$). When applying the hydrological threshold, we found the oxygen deficiency regions were mostly confined below pycnoclines and concentrated in summer. While adopting the biological threshold, the dead zone was larger especially in summer, and oxygen deficient water mass were also present above pycnoclines (Fig 2, 6). *A. tonsa* may have suffered from sub-lethal effects even when the DO is above 2 mg L$^{-1}$. For example, in the 2011-Summer cruise, less than a quarter (24%) of the vertical water column contained sufficient oxygen ($p$O$_2$ > $p_{crit}$) at the South, and more than half (63%) of the vertical water column could be lethal ($p$O$_2$ < $p_{leth}$) to *A. tonsa*, indicating a highly stressful and even lethal environment to copepods throughout most the water column. In other words, the conditions could be stressful to *A.tonsa* just below the surface layers (at 5m depth). If considering *A.tonsa*'s metabolic needs, the hypoxia threshold is around 4 mg L$^{-1}$ instead of 2 mg L$^{-1}$ in this case.

Although DO < 2 mg L$^{-1}$ is a commonly adopted standard in many estuary studies, this hydrological standard does not reflect the fact that oxygen solubility varies with temperature and salinity (Lange et al., 1972) and that different species have different tolerances toward oxygen deficiency (Diaz, 2001). Since oxygen solubility decreases with temperature and salinity, using DO < 2 mg L$^{-1}$ as a definition of hypoxia may underestimate the severity of oxygen deficiency in warm and salty ecosystems like the Gulf of Mexico. Compared with other seasonally hypoxic ecosystems, the Chesapeake Bay has moderate temperature and salinity, but it could be warm and salty in the southern region of the Bay during summer, where hypoxia is often overlooked compared to the northern region of the Bay. For example, in the 2011-Summer, our South station had similar DO as the North station at the same depth, but a much higher percentage of the vertical water column was biologically hypoxic (76% at South vs. 52% at North, Fig 2 & 6, more detailed analysis in Pierson et al., 2017) due to the higher salinity. As results, the South station sometimes provided less suitable habitat to *A. tonsa* than the North even though



the DO was similar or higher, and therefore the hypoxic habitat under warm and salty conditions were likely to be overlooked when using a fixed standard for defining hypoxia.

In addition to oxygen supply varying with temperature and salinity, oxygen demands also vary among species and stages. Usually, fast-swimming species and younger individuals require more oxygen at the same temperature and salinity than drifting species and older individuals (reviewed in Ekau et al., 2010). In this study, we compared the oxygen supply and demand of adult *A. tonsa* and concluded that the area and duration of biological hypoxia ($pO_2 < p_{crit}$) were larger and longer than hydrological hypoxia (DO < 2 mg L$^{-1}$). Meanwhile, gelatinous zooplankton are known to be better at tolerating hypoxia (Richardson et al., 2009). For example, the $p_{crit}$ of ctenophore *Mnemiopsis leidyi*, the dominant jellyfish in the main stream of Chesapeake Bay during summer, is 7 kPa At 25°C (Thuesen et al., 2005), which is almost half of the $p_{crit}$ of *A.tonsa* at the same temperature (13 kPa, Elliott et al., 2013), and thus the biologically defined hypoxia regions for *M. leidyi* would be smaller than the hydrologically defined hypoxia zone in the Bay. If applying the same philosophy, the biological hypoxia regions of large and fast swimming species like striped bass would be larger than the hydrological hypoxia zone in the Bay. Considering that response to low-dissolved oxygen is not universal, we recommended future research not only focus on dissolved oxygen but should also consider the effects of temperature, salinity, and species when evaluating the impact of hypoxia on ecosystems.

**4.2 Seasonal hypoxia and episodic hypoxia**

Species respond differently to hypoxia at various temporal scales. Under a permanent hypoxic ecosystem in the oxygen minimum zone, many organisms evolve physiological adaptations and genetic modifications to cope with the low dissolved oxygen environment through enhancing oxygen absorption (i.e. increase HbO2 affinity, increase gill surface area) and decreasing oxygen demands (i.e. decrease red cell ATP concentration) (Gracey et al., 2001; Mandic et al., 2009; Powers, 1980; Wood and Johansen, 1972). Many of the species in the oxygen minimum zone, particularly krill and myctophid fishes, turn hypoxic conditions to their advantage and take refuge from visually oriented predators during daytime (reviewed in Gilly et al., 2013). Species living in non-permanent hypoxic conditions on the other hand, mostly rely on behavioural adaptation or metabolic suppression to cope with temporally adverse conditions (reviewed in Childress and Seibel, 1998), and thus oxygen deficiency is more like a stressor than a refuge for organisms live under episodic/ seasonal hypoxic ecosystems like coastal dead zones.

The hypoxia in the Chesapeake Bay is mostly seasonal, especially pronounced in summer. Our PCA analysis indicated samples collected during summer were distinguished from water samples collected in spring as warmer, saltier, and having less dissolved oxygen in the bottom layers. Although the oxygen deficiency is temporary and localized, many studies have found negative effects of summer bottom hypoxia on species distributions (Breitburg, 1992; Keister et al., 2000), abundance (Roman et al., 1993), and diversity (Cooper and Brush, 1993). Previous studies found copepods show different diel migration patterns under seasonal hypoxia (Pierson et al., 2017), and their non-predatory mortality was higher in summer (Elliott et al., 2013a).

PCA results also indicated samples collected during the 2011 autumn cruise (September 21 − 26, 2011), right after Hurricane Irene (passed the Bay on Aug 27 − 28, 2011) and Tropical Strom Lee (passed the Bay September 7 − 11, 2011), were distinguished from the others (Fig 9). They were cooler, less salty, and less oxygenated compared with water samples collected from the previous year at the same location and season. Significant weather events like hurricanes and tropical storms could cause a hypoxia event by introducing a large amount of fresh water and organic matters. Palinkas et al. (2014) indicated that Hurricane Irene was a wind/ sediment resuspension event while Tropical Storm Lee was a hydrological/





sediment deposition event. The turbulence from Hurricane Irene was mostly confined in the tributaries, but the precipitation and sediment deposition from Tropical Storm Lee were mostly in the main stem of the Bay. Tropical Storm Lee brought heavy precipitation (22,002 m$^3$/ s) on Susquehanna River and resulted in the second highest recorded discharge behind the Tropical Storm Agnes in 1972 (Hirsch, 2012). As a result, the salinity of the 2011-Autumn cruises was much lower than the previous year. Some studies also observed short-term hypoxia in estuaries after hurricanes (Peierls et al., 2003; Stevens et al., 2006), and sometimes the recovery took months (Mallin et al., 1999). The effects of episodic hypoxia were less studied, and it was likely that we observed effects of the Hurricane Irene and Tropical Storm Lee during the 2011-autumn cruise. In this study we observed both seasonal and episodic hypoxia in this research, and more analysis will be conducted to understand their effects on zooplankton compositions and foodweb interactions.

## 5 Conclusion

Hypoxia in the Chesapeake Bay developed from the bottom water of the northern Bay in late spring, became established in summer, and dissipated in autumn. In general, water samples from the bottom water layers, during summer, and in 2011 contained less dissolved oxygen than samples from the surface layers, during spring, and in 2010. When applying a biological standard ($pO_2 < P_{crit}$ for *A. tonsa*), the oxygen deficiency area was larger and the event duration was longer than applying a fixed hydrological standard (DO < 2mg L$^{-1}$), therefore the severity of oxygen deficiency could be underestimated with a universal hydrological standard. The differences of hypoxia area and period according to a biological or a hydrological standard were especially pronounced in warm and salty conditions, like the southern Bay in summer. We concluded using oxygen concentration alone is not comprehensive to indicate the conditions of oxygen sufficiency of an ecosystem, instead we recommended considering temperature, salinity, and species differences when evaluating water quality and the effects of low dissolved oxygen on target species and ecosystems.

## 6 Data availability

Cruise details, deployment, and the measurements were uploaded to the Biological and Chemical Oceanography Data Management Office (DOI: 10.1575/1912/bco-dmo.687991; hyphen is part of the DOI), and the Scanfish and meteorology data from the six cruises were stored in the Rolling Deck to Repository (DOI: 10.7284/901570; 10.7284/901574; 10.7284/907638; 10.7284/901618; 10.7284/902443; 10.7284/902721).

## Acknowledgement

This study was supported by NSF Division of Ocean Sciences (NSF OCE-0961942). We especially thank chlorophyll data from Dr. Diane Stoecker, analytic advice from Dr. David Elliott, Stocker, and statistics consultation from Dr. Dong Liang at the Environmental Statistics Collaborative, University of Maryland Center For Environmental Science. We also appreciate hard work from the crew of the RV Hugh R. Sharp, Ginger Jahn, and Catherine Fitzgerald and advice from Dr. Mary Beth Decker, Dr. Edward Houde.

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





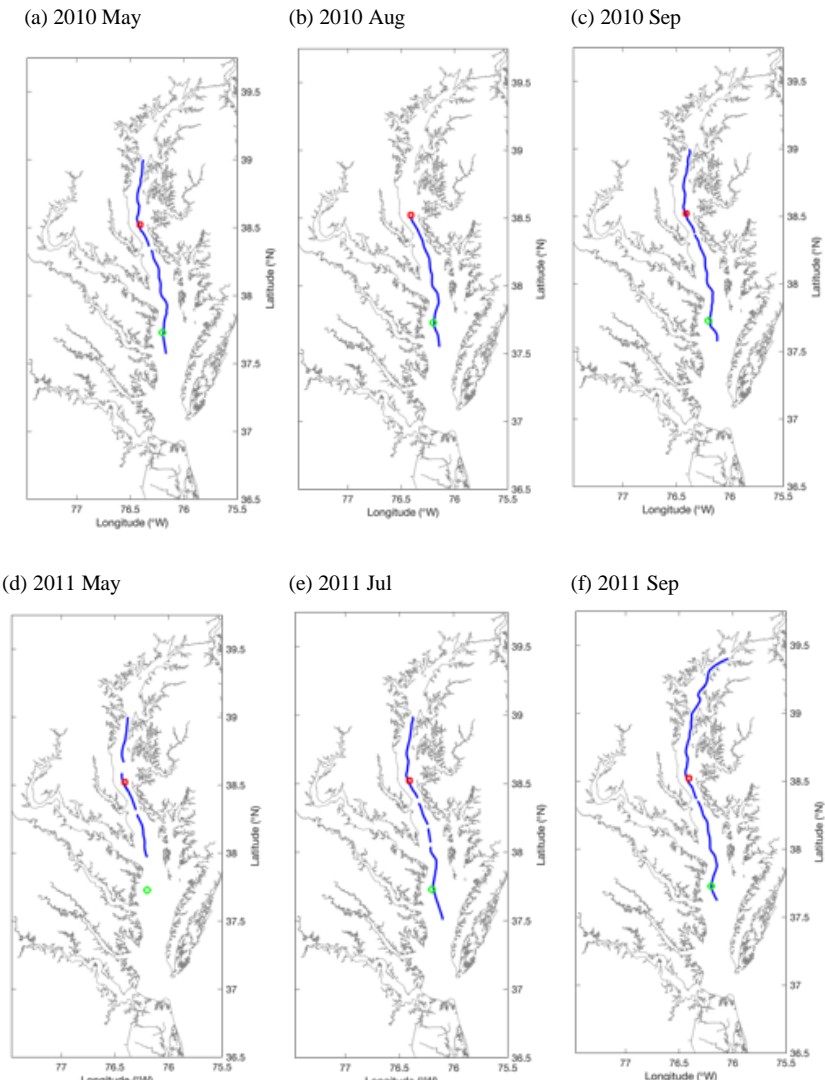

**Figure 1: The sampling maps of the Dead Zone Zooplankton research project. Red squares indicated the North station (38.528° N,76.418° W), green circles indicated the South station (37.738° N, 76.208° W), and the blue lines indicated the path of the Scanfish survey .**





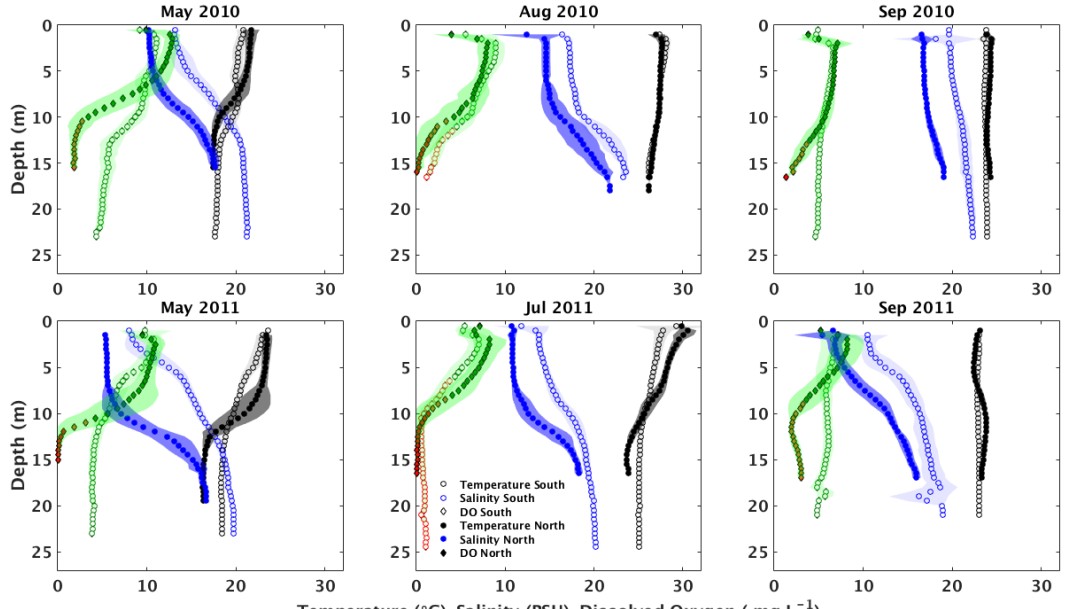

**Figure 2. Average temperature (black squares, °C), salinity (blue circles, PSU), and dissolved oxygen (multicolored diamonds, mg L$^{-1}$) from the CTD casts taken at North (closed) and South (open) stations during 2010 and 2011 cruises. Symbols represent mean values in 0.5 m bins from all CTD measurements at each depth, and shading indicated standard deviations (darker shading represents North and lighter shading represents South). Color filling of diamonds for dissolved oxygen represent partial pressure: above $P_{crit}$ (green), between $P_{crit}$ and $P_{leth}$ (orange), and below $P_{leth}$ (red). Modified from Pierson et al. (2017).**

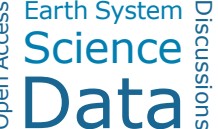



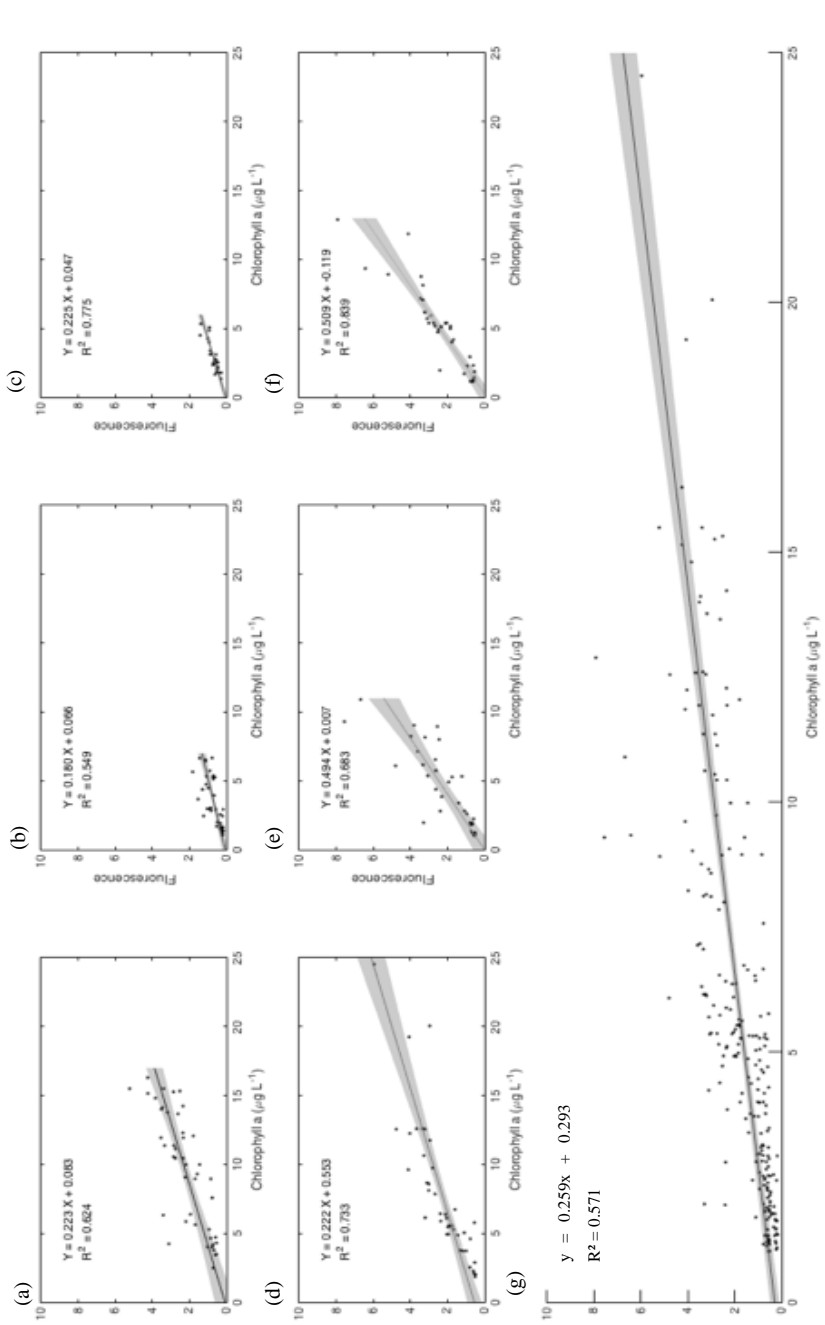

**Figure 3. The relationship between Chlorophyll-a and relative fluorescence collected from the 2010 May (a), August (b), September (c) and the 2011 May (d), July (e), September (f) cruise, and all cruises (g).**



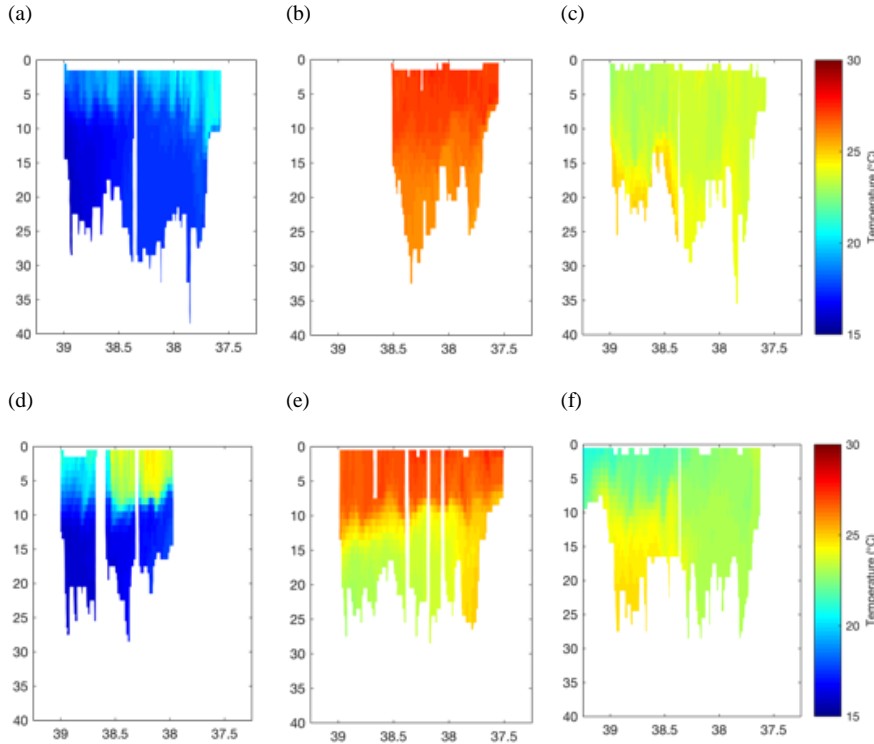

**Figure 4: The temperature (colorbar, C°) along the main channel of the Chesapeake Bay, collected from the Scanfish during the 2010 May (a), August (b), September (c) and the 2011 May (d), July (e), September (f) cruise.**



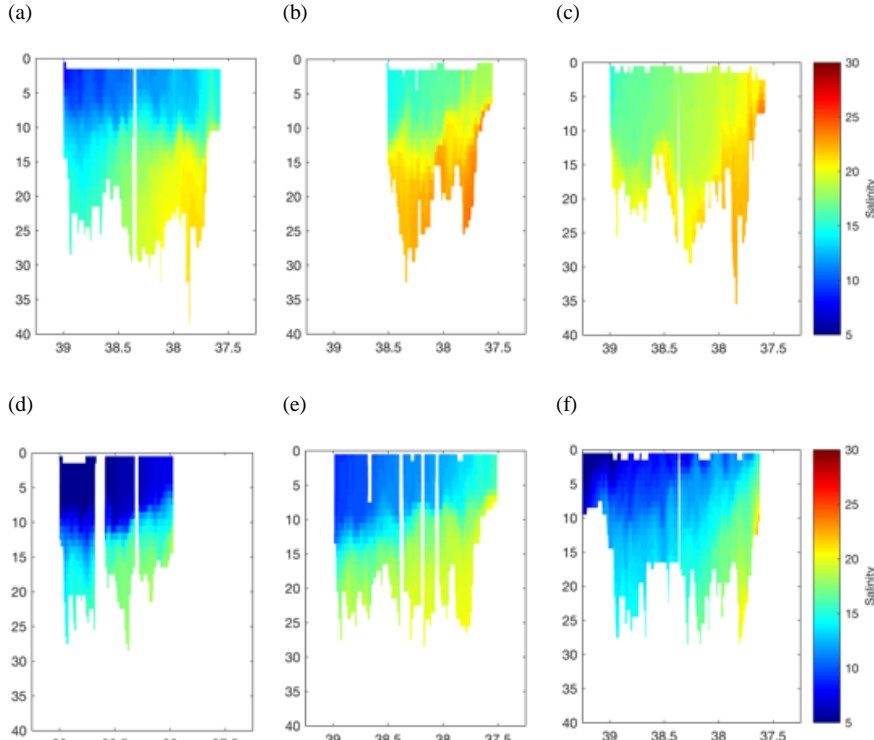

**Figure 5: The salinity (colorbar, PSU) along the main channel of the Chesapeake Bay, collected from the Scanfish during the 2010 May (a), August (b), September (c) and the 2011 May (d), July (e), September (f) cruise.**





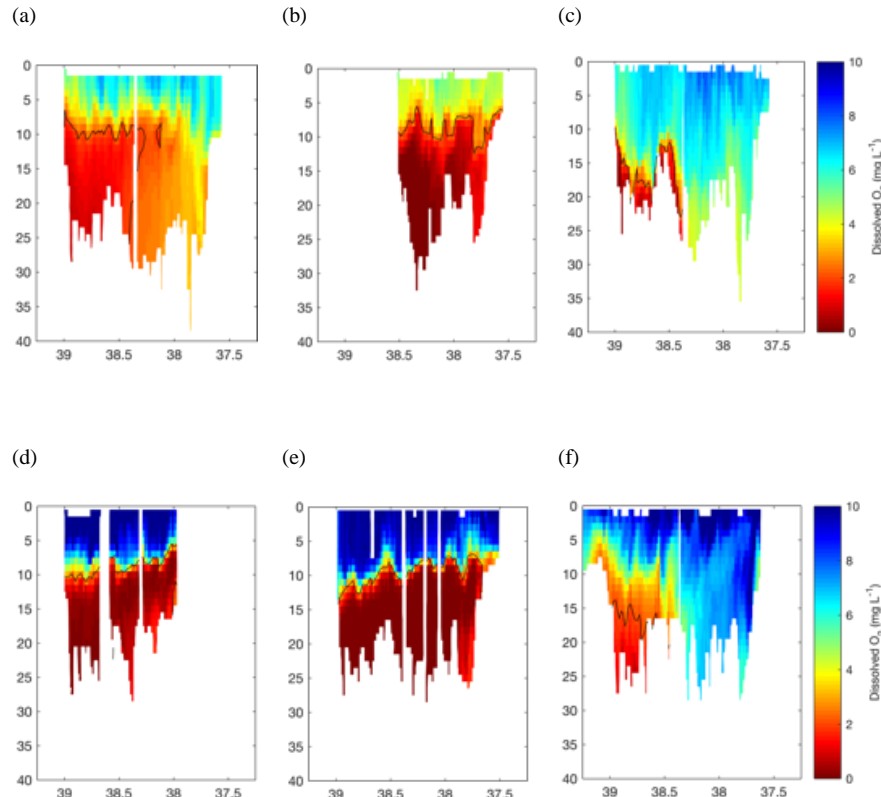

**Figure 6: The dissolved oxygen (colorbar, mg L$^{-1}$) along the main channel of the Chesapeake Bay, collected from the Scanfish during the 2010 May (a), August (b), September (c) and the 2011 May (d), July (e), September (f) cruise. Black lines indicated DO = 2 mg L$^{-1}$.**

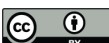

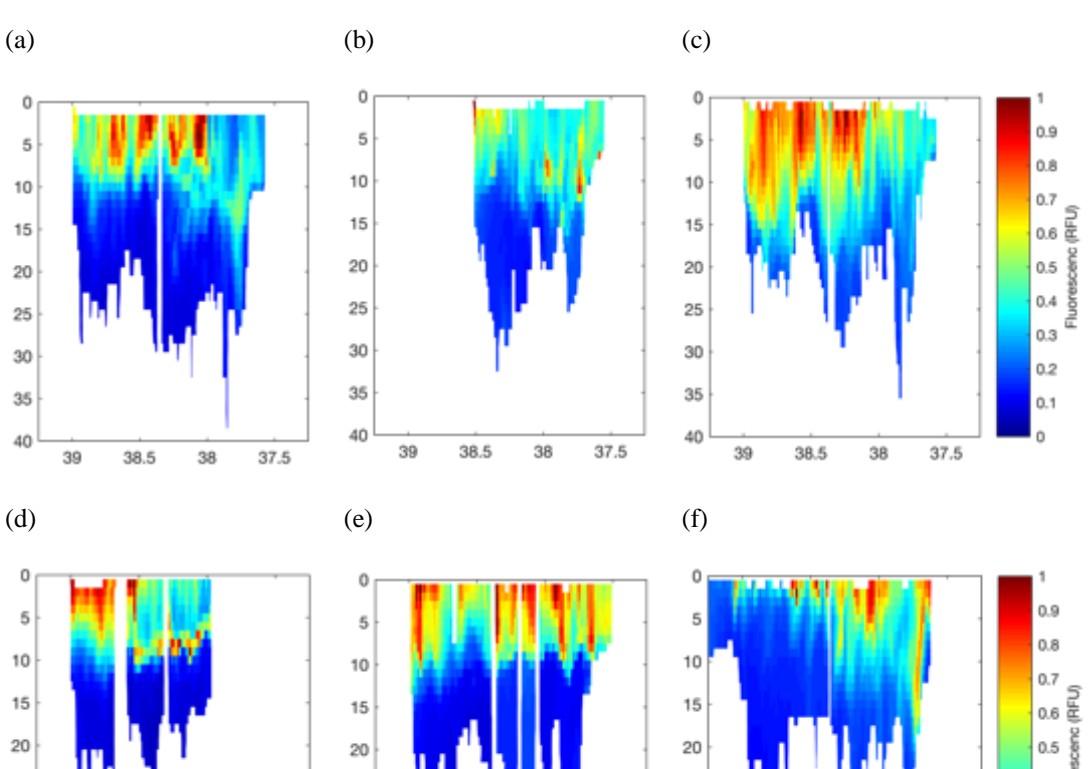

**Figure 7: The relative fluorescence (colorbar, RFU) along the main channel of the Chesapeake Bay, collected from the Scanfish during the 2010 May (a), August (b), September (c) and the 2011 May (d), July (e), September (f) cruise.**

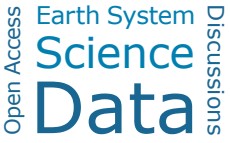

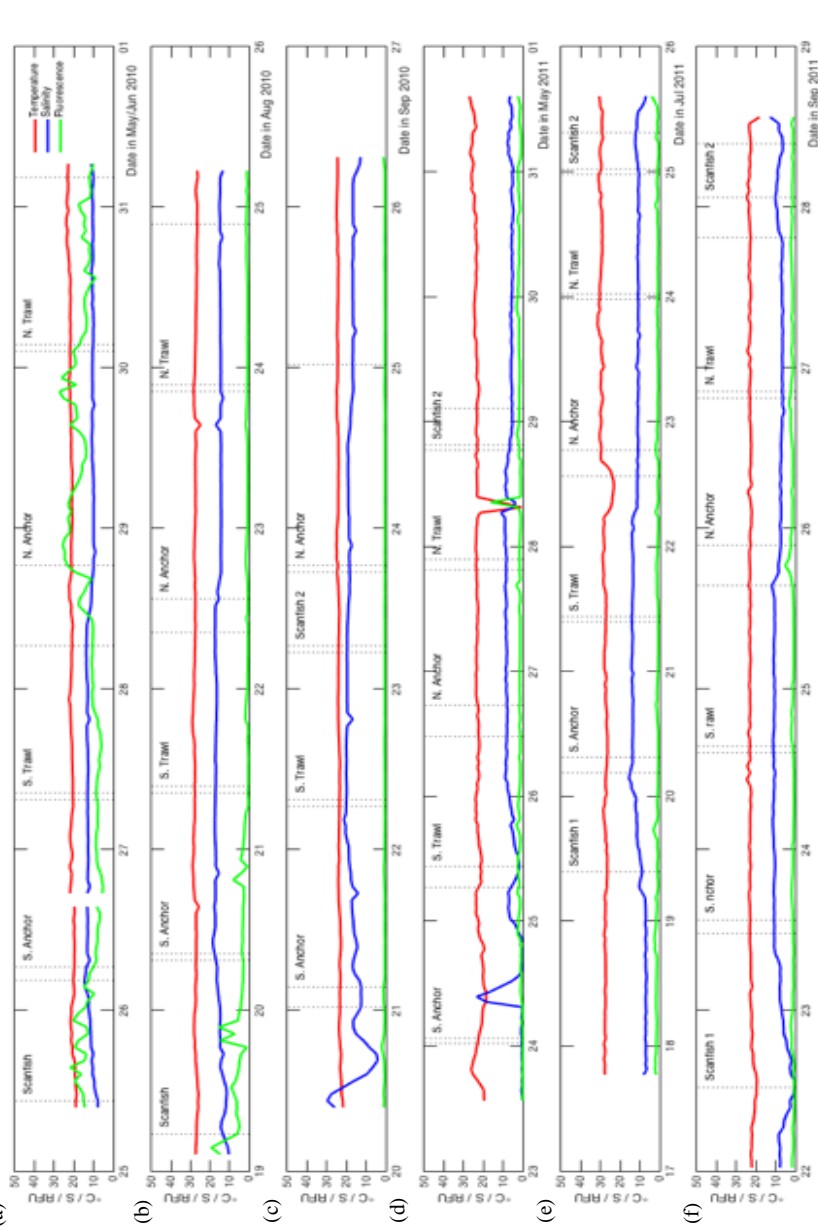

**Figure 8. The water surface temperature (red), salinity (blue), and fluorescence (green) along the main channel of the Chesapeake Bay collected SMS system of the R/V Sharp when conducting Scanfish (SF), southern anchor (SA), south trawl (ST), north anchor (NA), and north trawl (NT) during the 2010 May (a), August (b), September (c) and the 2011 May (d), July (e), September (f) cruise.**

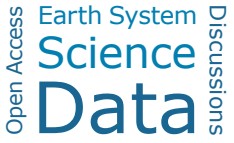

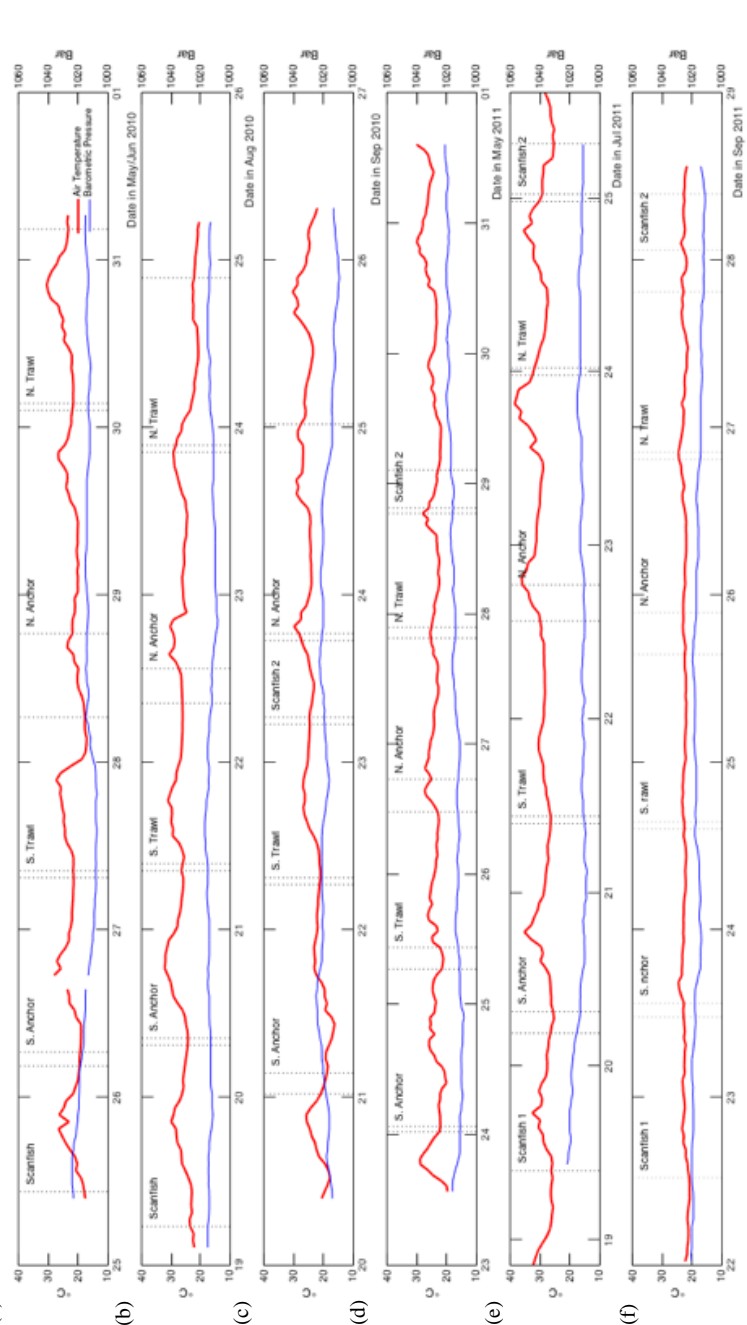

**Figure 9.** The air temperature (red, °C) and pressure (blue, bar) along the main channel of the Chesapeake Bay collected from the SMS system of the R/V Sharp when conducting Scanfish (SF), southern anchor (SA), south trawl (ST), north anchor (NA), and north trawl (NT) during the 2010 May (a), August (b), September (c) and the 2011 May (d), July (e), September (f) cruise.





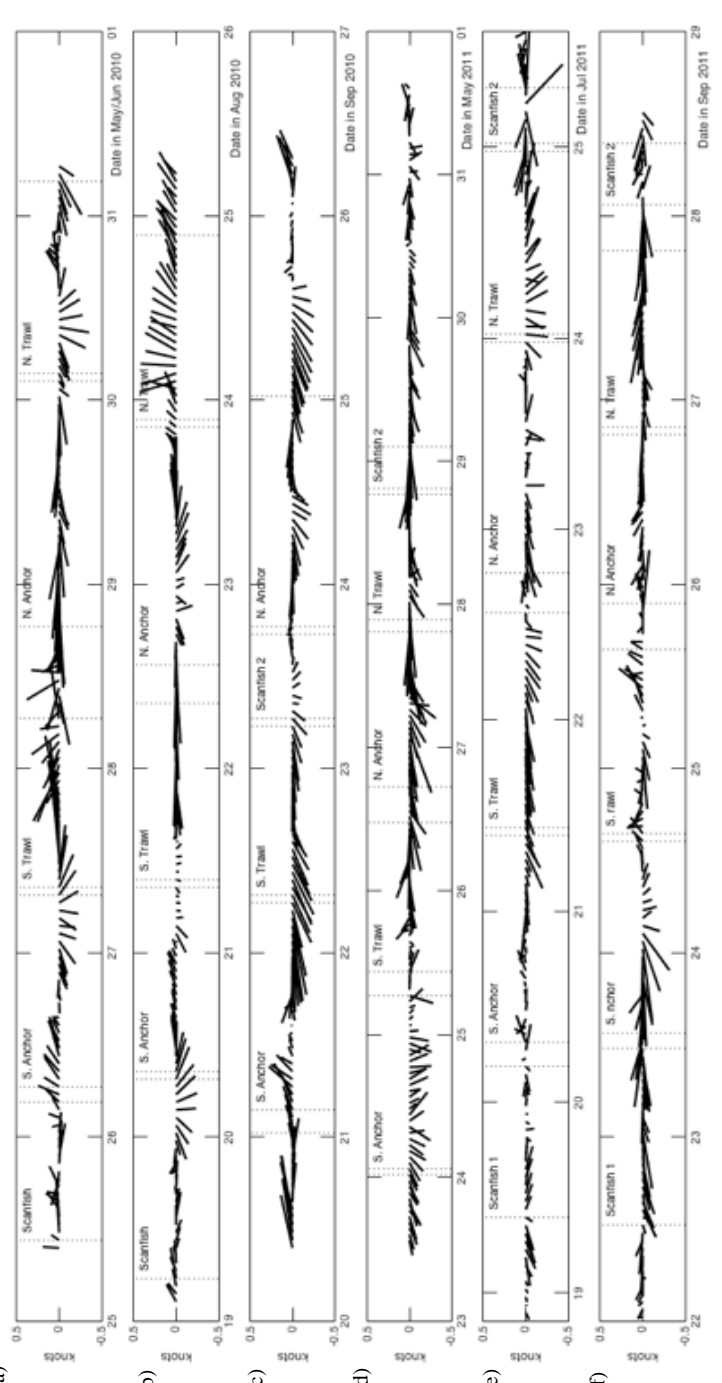

**Figure 10. True wind directions and speed (knots) along the main channel of the Chesapeake Bay collected from the SMS system when conducting Scanfish (SF), southern anchor (SA), south trawl (ST), north anchor (NA), and north trawl (NT) during the 2010 May (a), August (b), September (c) and the 2011 May (d), July (d), September (e), September (f) cruise.**



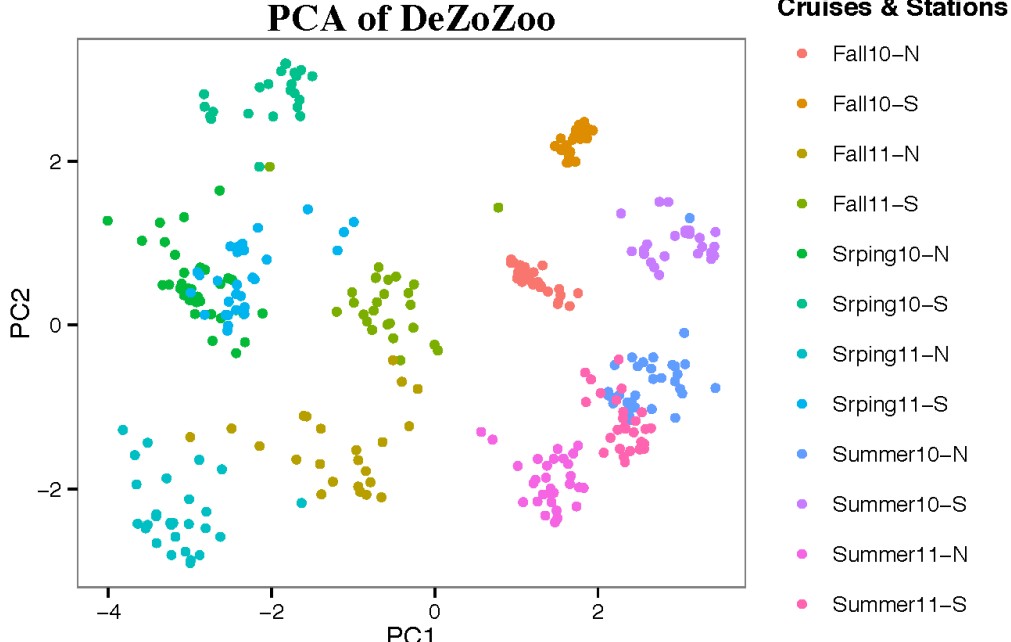

**Figure 11. Principle component analysis of average temperature, salinity, and dissolved oxygen of the water columns of above, at, and below pycnoclines from each CTD cast at the North and South anchor stations.**

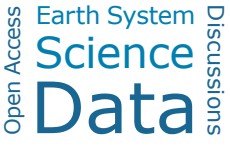

| Cruise | 2010 | 2011 |
|---|---|---|
| Season | | |
| Spring | 77 | 88 |
| Summer | 88 | 69 |
| Autumn | 64 | 66 |
| **Total Casts** | 229 | 223 |
| | | **452** |

Table 1. The numbers of CTD casts from each cruise.





(a)

|   | Eigenvalue | Difference | Proportion | Cumulative |
|---|---|---|---|---|
| 1 | 5.01 | 2.68 | 0.56 | 0.56 |
| 2 | 2.33 | 1.68 | 0.26 | **0.82** |
| 3 | 0.66 | 0.16 | 0.07 | 0.89 |
| 4 | 0.50 | 0.21 | 0.06 | 0.95 |
| 5 | 0.30 | 0.21 | 0.03 | 0.98 |
| 6 | 0.08 | 0.04 | 0.01 | 0.99 |
| 7 | 0.05 | 0.01 | 0.01 | 0.99 |
| 8 | 0.04 | 0.02 | 0.00 | 1.00 |
| 9 | - | 0.02 | 0.00 | 1.00 |

(b)

|   | Principal Component 1 | Principal Component 2 |
|---|---|---|
| DO above pycnoclines | -0.35 | 0.20 |
| DO at pycnoclines | -0.27 | 0.33 |
| DO below pycnoclines | -0.14 | **0.51** |
| Temp. above pycnoclines | **0.38** | -0.26 |
| Temp. at pycnoclines | **0.39** | -0.12 |
| Temp. below pycnoclines | **0.42** | -0.06 |
| Salinity above pycnoclines | 0.33 | **0.40** |
| Salinity at pycnoclines | 0.33 | 0.39 |
| Salinity below pycnoclines | 0.30 | **0.45** |

**Table 2. Eigenvalue (a) and eigenvectors (b) of principle component analysis of temperature, salinity, and dissolved oxygen (DO)**
5 **of the water columns from above, at, and below pycnocline from each CTD casts at both anchor stations.**

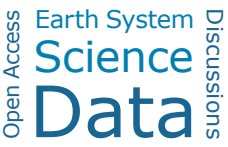

|   | LO | MO |
|---|---|---|
| C | ***2011-Spring (N)*** | 2010-Spring (N, S), 2011-Spring (S) |
| T | **2011- Autumn (N)** | 2011-Autumn (S) |
| W | ***2011-Summer (N, S), 2010-Summer (N)*** | **2010- Autumn (N, S), 2010-Summer (S)** |

**Table 3. Grouping cruises (Year-Season (Station)) according to the PCA results. All cruises were grouped into three temperature groups according to their PC1 scores (C = Cool, T = Temperate, W = Warm), and each group was divided into two subgroups according to their PC2 scores (LO = Less-oxygenated, MO = More-oxygenated). Underline letters indicated the averaged bottom DO < 2 mg L$^{-1}$, Bold letters indicated bottom $p$O$_2$ < $P_{crit}$, italic letters indicated $p$O$_2$ < $P_{leth}$.**

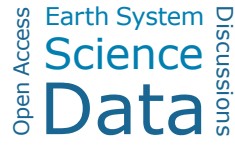

| Group | Sample Size | | Layer | Temperature | | Salinity | | $pO_2$ | | $P_{crit}$ | | $P_{leth}$ | |
| --- | --- | --- | --- | --- | --- | --- | --- | --- | --- | --- | --- | --- | --- |
| | LO | MO | | LO | MO | LO | MO | LO | MO | LO | MO | LO | MO |
| C | 28 | 87 | Surf. | 22.69 ± 0.53 | 21.19 ± 1.10 | 5.86 ± 0.40 | 11.56 ± 2.30 | 23.48 ± 2.32 | 26.79 ± 2.95 | 8.13 | 8.77 | 3.22 | 3.44 |
| | | | Pyc. | 19.95 ± 1.81 | 20.39 ± 1.55 | 8.06 ± 2.22 | 13.57 ± 2.92 | 13.80 ± 5.39 | 21.87 ± 5.06 | 8.26 | 8.74 | 3.26 | 3.43 |
| | | | Bot. | 17.18 ± 0.50 | 18.51 ± 0.75 | 12.76 ± 1.14 | 17.44 ± 1.75 | *__1.48__* ± *__1.36__* | 11.79 ± 4.33 | 7.88 | 8.27 | 3.13 | 3.27 |
| T | 23 | 26 | Surf. | 22.64 ± 0.23 | 22.94 ± 0.07 | 8.22 ± 1.12 | 11.22 ± 1.36 | 17.39 ± 3.93 | 17.14 ± 3.60 | 8.55 | 9.12 | 3.36 | 3.56 |
| | | | Pyc. | 23.14 ± 0.49 | 22.91 ± 0.07 | 11.06 ± 1.92 | 14.30 ± 1.52 | 16.92 ± 5.65 | 17.33 ± 4.17 | 9.13 | 9.62 | 3.57 | 3.74 |
| | | | Bot. | 23.61 ± 0.22 | 22.90 ± 0.06 | 13.61 ± 0.93 | 16.42 ± 0.79 | __7.36__ ± __1.68__ | 16.37 ± 1.36 | 9.73 | 9.95 | 3.78 | 3.85 |
| W | 87 | 86 | Surf. | 26.12 ± 1.44 | 26.27 ± 1.80 | 14.35 ± 2.27 | 17.50 ± 2.16 | 15.34 ± 2.82 | 19.18 ± 2.41 | 10.81 | 11.82 | 4.15 | 4.50 |
| | | | Pyc. | 24.87 ± 0.89 | 25.95 ± 1.62 | 15.99 ± 1.51 | 19.05 ± 1.94 | 11.82 ± 4.59 | 15.68 ± 3.74 | 10.72 | 12.09 | 4.12 | 4.60 |
| | | | Bot. | 24.39 ± 0.60 | 25.69 ± 1.34 | 17.85 ± 0.93 | 20.85 ± 1.53 | *__4.06__* ± *__4.34__* | __8.34__ ± __5.61__ | 10.91 | 12.41 | 4.19 | 4.71 |

**Table 4.** The Averaged (± S.D.) temperature, salinity, partial pressure of dissolved oxygen ($pO_2$) from three water layers (Surf. = above pycnoclines, Pyc. = at pycnoclines, Bot. = below pycnoclines), and the corresponding critical partial oxygen pressure ($P_{crit}$) and lethal oxygen partial pressure ($P_{leth}$) of each temperature (C = Cool, T = Temperate, and W = Warm) and dissolved oxygen subgroups (LO = less oxygenated, MO = more oxygenated). Sample size indicated the total numbers of CTD cast of each group. Bold $pO_2$ indicated $pO_2 < P_{crit}$ (Biological hypoxia), and bold and italic indicated $pO_2 < P_{leth}$.