# Peer review of "Evaluation oxygen deficiency in the Chesapeake Bay"

_Earth System Science Data, 2017_

## Referee Comment (RC2)

Review ESSD-2017-129, Chesapeake Bay Oxygen

I have read and strongly agree with comments from other reviewer. In contrast to that reviewer, I can access most of the data. But if one reviewer can and another can't, the authors clearly have an access problem. Once I do access the data, I encounter confusion and dis-array.

I can download and open the CTD data file and plot data from individual casts/profiles (e.g. reproduce something that resembles Figure 2). But all metadata about sensors, operations, quality control seems to come directly from SeaBird operators manuals. We get no information about calibration, data continuity, specific operations issues, etc. These authors have not shared with readers the real-world work of data quality control that makes data useful and sharable! Figure 2, already mentioned, evidently includes some shading intended to represent variability (among multiple casts during station occupation?) but nowhere in this text do the authors discuss variability or uncertainties. We might find some of this information in a prior publication, e.g. Pierson et al. 2017, but this reviewer like most ESSD readers wants necessary and useful metadata here, directly attached to and affiliated with this data.

I can also download and open the ScanFish and Safety Measurement System (ship meteorology) files from the R2R site but I confront 100s of .txt files. Evidently, according to the description on the landing page: "UNDERWAY DATA SETS (ORIGINAL FIELD DATA)". In other words, raw data? Individual .txt files have no headers, no metadata. The one I opened had at least half the fields filled with -99 values. No guidance to these files? Authors have not shared any quality control information nor any compiled products? We appreciate knowing that ship-owning institutions have properly archived raw data from all cruises but what value have the authors added? Nothing, evidently? If no value added, why submit to ESSD?

This R2R link - 10.7284/907638 - leads back to the BCO-DMO landing page and different CTD data file than the one listed earlier under a different doi (above).

Finding that discrepancy forced me to check all the six links:

| 10.7284/901570 | R2R | .txt files, raw data? | HRS100524JP |
|---|---|---|---|
| 10.7284/901574 | R2R | .txt files, raw data? | HRS100819JP |
| 10.7284/907638 | BCO-DMO via R2R | ftp site with 6 files | different set of CTD data? HRS100920JP? |
| 10.7284/901618 | R2R | .txt files, raw data? | HRS110525JP |
| 10.7284/902443 | R2R also NCEI for ADCP data? | .txt files, raw data? | HRS110719JP |
| 10.7284/902721 | RSR | .txt files, raw data? | HRS110922JP |

Following links for multiplexed (primary) UDel SMS files - not because the authors have provided such guidance but because on the landing pages each doi leads us to, those represent the only download-able data - a user finds five similar files, except that one apparently has additional ADCP and CTD data submitted to NCEI. One of the doi links that we expect to lead to R2R instead leads back to BCO-DMO and a new (to us) set of CTD data.

With these files downloaded to my computer, I confront a labelling disaster. No consistent codes, no obvious dates or locations, evidently hundreds of machine-generated file and folder names. No master directory. For my own usage, and if the manuscript convinced me of the quality and validity of these data - which unfortunately it does not - I might spend hours and days trying to compile and make sense of these data. In their present disarray and undocumented form, I would not share these data with an undergraduate oceanography class nor with an oceanography graduate student; not fair to them to confront a data morass that I had not penetrated and solved. ESSD exists to share data, to facilitate data exchange, encourage data quality, etc. This long confusing array of mostly ill-described and undocumented raw data does not fit the ESSD model.

I understand that authors need to fit with and work with formats and policies imposed by R2R and BCO-DMO, but we get no evidence here that authors pursued quality beyond the standard institutional checks or made any effort to compile data in a friendly informative manner for subsequent users.  We need a few (two or three) master files, of compiled, cross-checked, quality-controlled data, products that demonstrate authors skills and efforts.  If R2R or BCO-DMO do not want to support such compiled products, Pangaea or Zenodo will both feel pleased to provide excellent archive and access services.

I echo comments of the other reviewer: this reads like a research paper, not like a data publication.  We get no details about oxygen sensors, fluorimeters, repeatability, calibration, maintenance, etc.  Please can the authors look at other ESSD papers, e.g. https://doi.org/10.5194/essd-9-861-2017 for biochemical ARGO data and any of several other papers currently in review in ESSD on seawater nutrients, freshwater quality (and BOD), hydrography and nutrients, etc. to understand formats, organisation, and quality control presentations of data sets as expected by ESSD.

To reproduce Figures 4 through 7, users need to start again from raw .txt files?  Authors need to at least leave us a trail to know what tools they used.  Better, they need to share their compiled data sets!

This reader does not extract informative or useful information from Figures 8 through 10.  They remind this old oceanographer of watching data emerge on a strip chart recorder, to confirm instruments working properly but not as the basis of systematic analysis.  The authors seem to use these plots simply to show us they collected data on such-and-such date along the axis of the Bay?

Because the authors have provided this reader so little basis for confidence in their measurements and such limited access to their compiled data, I have no confidence in their PCA.  Most of us could sketch a similar outcome - north different to south, spring different to fall, one year different to the next - for any region based on no data whatsoever.  Again I confront this question: what have the authors done here in terms of quality control and compilation to make this data particularly useful for sharing and subsequent analysis?  I don't doubt that these authors have taken those necessary steps but they have not shared with us as readers and users.  Very many researchers might take an interest in this work and the underlying data if available; very few researchers have time to deal with hundreds of raw .txt files.

Reject in present format, authors need to decide whether they want to share useful data through ESSD or write a research paper about Chesapeake oxygen.  Both, perhaps?  They haven't provided sufficient access, metadata and quality control to provide a useful product via ESSD.

---

## Referee Comment (RC1) · Anonymous Referee #1 · 3 Mar 2018

The authors present the results of hydrographic surveys carried out in Chesapeake Bay in 2010 and 2011. They assess the hypoxic state of the water column relative to a hydrological standard of 2 mg l-1. They also calculate a biological standard based on A tonsa and assess the water column against this standard. Whilst it is an interesting study, I do not think that it can be published in its current state due to insufficient detail in the methods section.

Numbered comments below refer to the guidance for reviewers given on the ESSD website.

1. Read the manuscript

Whilst some of the data have been presented in Elliot et al (2013a) and Pierson et

al (2017) this paper does provide doi links to the datasets and therefore makes them publicly accessible. The data would be of use in the future for researchers to compare changes in state in the Bay. There are insufficient details given in the methods to allow other researchers to re-use the data. Within the manuscript there is no information about the instrumentation and sensors which are used on the CTD, Scanfish and SMS systems although the CTD Dataset description file does give details of instruments and serial numbers. There are no details about which parameters are measured on the SMS and Scanfish packages, no information about calibration regimes, logging regimes or uncertainty for each parameter. Figure 3 presents chlorophyll a data but there is no mention in the methods about samples being collected and analysed for chlorophyll a. It is not possible to assess the data quality as there is insufficient information given. The aims of the manuscript focus on the science rather than a comprehensive presentation of the datasets. There is insufficient detail in the methods section for other researchers to assess the quality of the data and therefore re-use the data.

2. Check the data quality

I was able to access the CTD data and download them using the doi. When I click on the doi links for the Scanfish and SMS system data, I get to a description of the underway datasets and a list of file names but could not download the actual data. As discussed above there is no discussion of uncertainties or calibrations.

3. Check the presentation quality

The CTD data and metadata are usable in current format although it is not possible to make an assessment of data quality. I couldn't assess the other data as I could not access it.

4. Check the publication

The analytical section of the paper is much more comprehensive than the methods section. Why have the authors chosen ESSD given the focus of the manuscript is on

the assessment of hypoxia in Chesapeake Bay rather than on a full description of the datasets?

Significance – the data are unique and potentially useful if sufficient meta data are included.

Data quality – unable to assess due to insufficient details in the manuscript.

Presentation quality – I have some specific comments in the separate annotated pdf.

Please also note the supplement to this comment:
https://www.earth-syst-sci-data-discuss.net/essd-2017-129/essd-2017-129-RC1-supplement.pdf